# Could Vitamins Have a Positive Impact on the Treatment of Parkinson’s Disease?

**DOI:** 10.3390/brainsci13020272

**Published:** 2023-02-06

**Authors:** Manas Ranjan Sahu, Linchi Rani, Arun S. Kharat, Amal Chandra Mondal

**Affiliations:** 1Laboratory of Cellular & Molecular Neurobiology, School of Life Sciences, Jawaharlal Nehru University, New Delhi 110067, India; 2Laboratory of Applied Microbiology, School of Life Sciences, Jawaharlal Nehru University, New Delhi 110067, India

**Keywords:** Parkinson’s disease, oxidative stress, neurodegeneration, antioxidant, vitamin A, vitamin B, vitamin C, vitamin D, vitamin E, vitamin K

## Abstract

Parkinson’s disease (PD) is the second most common progressive neurodegenerative disorder after Alzheimer’s disease. Pathophysiologically, it is characterized by intracytoplasmic aggregates of α-synuclein protein in the Lewy body and loss of dopaminergic neurons from substantia nigra pars compacta and striatum regions of the brain. Although the exact mechanism of neurodegeneration is not fully elucidated, it has been reported that environmental toxins such as MPTP, rotenone, paraquat, and MPP^+^ induce oxidative stress, which is one of the causative factors for it. To date, there is no complete cure. However, the indispensable role of oxidative stress in mediating PD indicates that antioxidant therapy could be a possible therapeutic strategy against the disease. The deficiency of vitamins has been extensively co-related to PD. Dietary supplementation of vitamins with antioxidant, anti-inflammatory, anti-apoptotic, and free radical scavenging properties could be the potential neuroprotective therapeutic strategy. This review summarizes the studies that evaluated the role of vitamins (A, B, C, D, E, and K) in PD. It will guide future studies in understanding the potential therapeutic role of vitamins in disease pathophysiology and may provide a framework for designing treatment strategies against the disease.

## 1. Introduction

Parkinson’s disease (PD) is a progressive neurological disorder characterized by α-synucleinopathy in the form of Lewy bodies (LB) and selective loss of dopaminergic neurons (DA-ergic) in the brain [1]. Globally, more than 1% of the population above 60 is affected by this disease [2]. PD patients exhibit some motor (bradykinesia, rigidity, resting tremor, postural instability) and some non-motor (dementia, sensory abnormalities, sleep disorders, and autonomic dysfunction) symptoms. The exact etiology of PD remains unknown. However, in-vitro and in-vivo studies have identified some neurotoxins such as 1-Methyl-4-phenyl-1, 2, 3, 6-tetrahydrodropyridine (MPTP), 1-methyl-4-phenylpyridinium (MPP^+^), 6 Hydroxy dopamine (6-OHDA), and paraquat (PQ) as inducers of Parkinsonian features, including LB formation. The formation of LB in the brain gives rise to oxidative stress, mitochondrial dysfunction, neuroinflammation, and aberrant cellular apoptosis, thus causing DA-ergic neurodegeneration [3]. The etiopathogenesis of PD is widely attributed to neuroinflammation. Chronic neuroinflammation damages the blood-brain barrier (BBB), providing an inlet to peripheral immune cells, which leads to CNS infiltration. Moreover, activated astrocytes and microglia release proinflammatory cytokines, including IL-6, TNF-α, and IL-1β in an uncontrolled manner, leading to disease progression by inducing neuroinflammation [4]. Additionally, neuroinflammation is also associated with the pathogenesis of other Parkinsonian-related syndromes, including progressive supranuclear palsy (PSP)-Richardson syndrome (PSP-RS), corticobasal syndrome (CBS), and multiple system atrophy-Parkinsonism (MSA-P). Neutrophil-to-lymphocyte ratio (NLR) and platelet-to-lymphocyte ratio (PLR) are attributes of peripheral inflammatory responses in atypical Parkinsonism. The clinical studies have shown that NLR and PLR are significantly higher in PD and MSA-P as compared to the control group. However, in the case of PSP-RS, only NLR was significantly higher, but in CBS, increase in NLR was non-significant compared to the control group [5,6]. This suggests that NLR could be considered for the diagnosis of atypical Parkinsonism. Nowadays, dietary supplementation of vitamins having antioxidant, anti-inflammatory, and free radical scavenging properties are becoming popular as a primary neuroprotective therapeutic strategy against PD [7]. In this review, we have comprehensively summarized the therapeutic role of various vitamins in PD, along with the possible molecular mechanisms involved.

## 2. Oxidative Stress and Neuroinflammation at the Core of PD Pathogenesis

Oxidative stress is one of the prime factors for DA-ergic neurodegeneration, as evident in numerous clinical studies [8]. A higher level of oxidized proteins, lipids, and DNA has been detected in the brain tissue of both sporadic and familial PD patients [9]. An elevated level of reactive oxygen/nitrogen species (ROS/RNS) and decreased activity of antioxidants such as glutathione peroxidase, catalase (CAT), and superoxide dismutase (SOD) has also been reported in PD patients [9,10]. The genetic mutations in familial PD cases are closely associated with inducing oxidative stress during PD pathogenesis. Mutations in the gene synuclein alpha (SCNA), which leads to the over-production of the protein α-synuclein, have been shown to trigger LB formation via ROS production. Similarly, mutations in some of the essential genes associated with mitochondrial membrane integrity and functionality, such as phosphatase and tensin homolog (PTEN)-induced kinase 1 (PINK1), parkin (PRKN), and DJ-1, have also been reported to cause early onset PD by increasing the cellular oxidative stress burden [11,12]. Findings from epidemiological studies state that sporadic PD cases arising from exposure to agrochemicals such as PQ, MPTP, rotenone, maneb, and other environmental toxins cause nigral toxicity primarily due to oxidative stress in nigral DA-ergic neurodegeneration [10,13]. On the same level, mounting evidence shed light on the role of neuroinflammation in the pathogenesis of PD, highlighting increased activation of astrocytes and microglia. LB formation activates microglia, which also significantly contribute to oxidative stress and can produce proinflammatory cytokines, such as IL-6, IL-1β, and TNF-α, triggering DA-ergic neurodegeneration. Proinflammatory cytokines in CSF and brain and increased activated microglia in the substantia nigra pars compacta (SNpc) have been detected in PD patients. In the end, inflammatory peripheral immune cells reach to brain via BBB, and further increase neuroinflammation-mediated neurodegeneration [11,14]. The pathogenesis of PD initiates from gut and dietary interventions may alter the gut microbiota, resulting in the modulation of the inflammatory response, a risk factor for PD. The studies have shown that intake of probiotics (fermented milk having Lactobacillus casei Shirota) significantly improved bowel habits and stool consistency in PD patients, suggesting that probiotics could rebalance the gut microbiota-associated change in PD, thereby reducing the neuroinflammation [15]. Other food supplements—fruits, vegetables, and spices—possess natural compounds, vitamins, phytochemicals, and minerals that can delay or ameliorate the pathological symptoms of neurodegenerative disease, improve learning, improve cognitive function, and improve wellbeing [14]. The phytochemicals, such as flavonoids commonly found in orange, apple, strawberry, tea, cocoa, and red wine with antioxidants and anti-inflammatory properties, are associated with lower risk of PD. The studies have shown that flavonoids inhibited α-synuclein aggregation, protected DA-ergic neurons from apoptosis and oxidative damage. Caffeine with anti-inflammatory properties, is a well known antagonist of adenosine-2A receptors and is associated with reduced risk of PD. The in vivo studies have shown that caffeine inhibits microglia activation resulting in amelioration of neuroinflammation [15]. However, dairy products, such as milk, are associated with higher incidence of PD. Casein present in the milk decreases uric acid levels in the serum of individuals. High serum uric acid may decrease risk and duration of PD [16]. These shreds of evidence suggest that a balanced healthy nutrition may have neuroprotective effect. Thus, oxidative stress and neuroinflammation forms a mainstay mechanism during PD pathogenesis, which can be targeted for developing an effective therapeutic option against PD. Since vitamins are bioactive compounds rich in antioxidative properties, their application in combating an oxidative stress-driven diseases such as PD holds immense therapeutic value.

## 3. Role of Vitamins in the Pathogenesis of PD

Vitamins are well known for their anti-inflammatory and antioxidant properties. Mounting evidence has shed light on how a deficiency of various vitamins increases the risk of PD onset [7,17,18,19,20]. Since depletion in antioxidant enzymes and enhanced oxidative stress leads to the onset and progression of PD pathogenesis, vitamin supplementation can be valuable in the therapeutic strategy against PD. A brief account of various vitamins (A, B, C, D, E, and K) and their therapeutic implications in PD have been described in the following sections.

### 3.1. Vitamin A

Vitamin A (retinol) is a major antioxidant in the diet and is abundant in fish, meat, dairy products, and plants. During the development of the central nervous system, vitamin A regulates the expression of genes involved in brain development and controls neural tube patterning and neuronal differentiation [21]. Several studies have shown the neuroprotective effects of vitamin A in PD. In vitro studies have shown that retinoic acid, the biologically active form of vitamin A, protects against 6-OHDA and 1-methyl-4-phenylpyridinium (MPP^+^)-induced neurotoxicity via activating protein kinase B (Akt) signaling, decreasing p53 levels and increasing Bcl-2 activation [22]. It has been demonstrated that the administration of retinoic acid reduced PD-related motor impairment and elevated dopamine (DA) levels. Furthermore, it prevented the loss of DA-ergic neurons in several rat PD models [22]. Vitamin A and its derivatives function by activating retinoid receptors, specifically the retinoic acid receptor (RAR) and retinoid X receptor (RXR). These receptors, in turn, inhibit the activation of Nur77, a pro-apoptotic protein, thus protecting DA-ergic neurodegeneration [22,23]. Furthermore, vitamin A treatment lowers the serum levels of various proinflammatory cytokines, such as TNF-α, Interleukin (IL)-1β, and Iba-1, alleviating neuroinflammation in the 6-OHDA-induced PD model [3]. The immunomodulatory effect of retinoic acid is carried out via inactivation of receptor-advanced glycation-end (RAGE) products, key regulators of p38 mitogen-activated protein kinase/nuclear factor kappa B (p38MAPK/NF-κB)-associated inflammatory cytokine production during PD [23]. Contrary to the above findings, some studies have also shown that vitamin A promotes oxidative stress, resulting in cell death. Treatment of neuronal cells with vitamin A, at a concentration above the cellular physiologically available concentration, elevated α-synuclein phosphorylation, and increased oxidative stress level leading to progression of neuronal death [1]. In line with this, oral supplementation of vitamin A (retinyl palmitate, 3000 IU/kg/day) for 28 days also failed to prevent DA-ergic neurodegeneration in the SNpc region of the brain in a 6-OHDA-induced PD rat model [3]. On the other hand, a cohort study involving 42 PD patients and 42 healthy controls found no correlation between the serum levels of vitamin A, its receptor, retinol-binding protein, and PD progression [24]. Similarly, another cohort study involving 63,257 men and women aged 45–74 from the Singapore Chinese population also failed to establish any dose-dependent correlation of dietary vitamin A with the risk of developing PD [25]. Despite the crucial role of vitamin A in the development and functioning of the nervous system, the defining role of vitamin A in PD pathogenesis remains ambiguous. Thus, more studies are needed to understand the implications of vitamin A deficiency and supplementation on PD pathogenesis. Figure 1 provides a brief overview of the potential mechanistic significance of vitamin A in attenuating pathophysiological alterations related to PD.

### 3.2. Vitamin B Family

Vitamin B family is a family of water-soluble vitamins. Numerous studies have shown that its family members have antioxidant and anti-inflammatory effects and thus serve as a neuroprotective against multifactorial neurodegenerative disorders, including PD.

#### 3.2.1. Vitamin B_1_

Vitamin B_1_ (thiamine) is commonly found in organ meat, egg, fish, lean pork, beef, legumes, wheat germ and whole grain, and nuts [26]. Thiamine uptake occurs in the body’s small intestine by two transporters named THTR1 and THTR2. Thiamine deficiency (TD) is associated with an increased risk of PD. In PD patients, there is a decrease in the level of α-ketoglutarate dehydrogenase complex (an enzyme associated with TD) in the SNpc region of the brain. This reduction is correlated with degeneration severity [26]. Patients suffering from Parkinsonism dementia have shown a decrease in the activity of thiamine triphosphate (TPP), a metabolically active form of thiamine, in the frontal cortex region of the brain [27]. Genetic alterations in thiamine metabolism also cause neurological diseases such as PD, which can be treated with a high dose of vitamin B_1_ [28]. The low dietary vitamin B_1_ intake in two to eight years before diagnosing PD is associated with olfactory dysfunction, a non-motor symptom related to increased risk of PD [29]. These findings suggest that deficiency of vitamin B1 promotes neuronal death, leading to increased risk for PD, and supplementation of the same may ameliorate pathological changes associated with PD. A clinical study on PD patients found that high doses of vitamin B_1_ improve motor symptoms from 31.3% to 77.3% of the Unified Parkinson’s Disease Rating Scale (UPDRS) in PD patients receiving no other anti-Parkinson therapy [28]. It has also been demonstrated that elevated plasma thiamine is associated with a reduced risk of mild cognitive impairment (MCI) in male PD patients [30]. Clinical studies have shown a significant improvement in the motor and non-motor symptoms in PD patients when administered daily 100–200 mg doses of thiamine intramuscularly [28]. Mechanistically, the PD ameliorative effects of B_1_ are mediated through the regulation of apoptotic signaling pathway involving the anti-apoptotic protein Bcl-2 and the pro-apoptotic protein p53 in the neuronal cells. Further, vitamin B_1_ also inhibits the GSK-3β activation associated neuroinflammatory responses, including activation of astrocytes in the PD brain [27]. Though significant studies have highlighted the beneficial role of vitamin B_1_ supplementation in the management of PD, the outcomes of the clinical study are inconsistent. Therefore, careful re-evaluation of vitamin B_1_ doses and combination of nutrients and the route of administration should be considered in the clinical studies [26]. Though regular dietary intake of vitamin B_1_ may reduce the risk of PD, research is needed to elucidate the therapeutic potential of vitamin B_1_ against PD.

#### 3.2.2. Vitamin B_3_

Niacin, or vitamin B_3_, is commonly present in foods including fish, meat, vegetables, and wheat [31]. Physiologically, it causes neural progenitors to differentiate into serotonergic and DA-ergic neurons [32]. The biological activity of vitamin B_3_ is mediated through the activation of a GPCR protein, GPR109A. Vitamin B_3_ supplementation has been shown to alleviate various symptoms associated with PD. In a Drosophila PD model, it has been shown that dietary supplementation with high doses of nicotinamide, an active form of vitamin B_3_, suppresses mitochondrial abnormalities and improves PD-associated motor deficits [33,34]. Additionally, it has also been shown that nicotinamide supplementation can prevent DA depletion and DA-ergic cell death in the SNpc region of the brain [35]. Further, clinical findings have revealed that vitamin B_3_ supplementation ameliorates neuroinflammation by modulating macrophage polarization from M1 (pro-inflammatory) to M2 (counter-inflammatory) in PD patients. Mechanistically, vitamin B_3_ promotes the biosynthesis of the classical enzyme cofactor nicotinamide adenine dinucleotide (NAD) and mediates the release of nicotinamide by poly-ADP ribosylation. This generates an anti-inflammatory response, which alleviates DA-ergic neurodegeneration caused by neuroinflammation [4,36,37]. These findings suggest that dietary supplementation with vitamin B_3_ may ameliorates oxidative stress and neuroinflammation, which would therefore prevent the death of DA-ergic neurons.

#### 3.2.3. Vitamin B_6_

Vitamin B_6_ can either be obtained from a dietary source or synthesized by the microbiota of the human large intestine. Vitamin B_6_ and its metabolites play a vital role in different metabolic processes, including antioxidant effect, neurotransmitter and amino acid metabolism, synthesis of protein and polyamines, metabolism of lipids and carbohydrates, erythropoiesis, and mitochondrial function [38]. Multifactorial neurological disorders such as PD, Alzheimer’s disease (AD), autism, schizophrenia, and epilepsy are associated with intracellular deficiency of pyridoxal 5′-phosphate, the active form of vitamin B_6_ in the liver [39]. A single-cell whole genome expression profiling study from human SNpc in PD patients has observed a genetic variation in the pyridoxal kinase (PDXK) gene, which is involved in the metabolism of vitamin B_6_/DA, is associated with an increased risk of PD [40]. A deficiency of vitamin B_6_ causes status epilepticus and early-onset epilepsy in PD patients [41]. A case-controlled study in Japan involving 249 PD patients and 368 healthy controls reported that low dietary intake of vitamin B_6_ is correlated with an increased risk of PD [42]. In agreement with these findings, a population-based cohort study on 5289 participants over the age of 55 years in Rotterdam by L M L de Lau et al. found that high dietary intake of vitamin B_6_ was associated with a significantly reduced risk of PD [19]. These pieces of evidence indicate that dietary intake of vitamin B_6_ is indispensable for health and may have a neuroprotective effect against PD. However, further studies are needed to give more depth into the mechanism involved.

#### 3.2.4. Vitamin B_12_

Vitamin B_12_ (cobalamin) commonly found in milk products, meat, and fish is one of the essential micronutrients which plays a crucial role in growth, nervous system, cognition, and chronic brain disorders [43]. Idiopathic PD (IPD) patients with hyperhomocysteinemia were observed to be deficient in vitamin B_12_, suggesting its association with Parkinsonian disorders [44]. Several studies have attempted to determine the relationship between vitamin B_12_ and PD. It has been shown that deficiency of vitamin B_12_ in PD patients is associated with cognitive impairment and gait impairment, rapid progression of disease, neuropathy, and rapid worsening of ambulatory capacity [45]. A population based-cohort study observed that PD patients with higher baseline level of vitamin B_12_ at PD diagnosis were associated with reduced risk of dementia [46]. A three-year longitudinal cohort study with 1741 individuals indicated that people receiving multivitamin (MVI) and B_12_ + MVI had a reduced hazard ratio for UPDRS than those receiving no supplement [47]. According to studies, vitamin B_12_ deficiency can exacerbate apoptosis and cause Parkinsonian phenotypes in rats by impairing the synthesis of S-adenosylmethionine. Mechanistically, vitamin B_12_ enters the cell via CD320 receptor-mediated transport, reduces oxidative stress, eases movement disorder, and restores mitochondrial function, preventing the degeneration of DA-ergic neurons. [48,49]. Furthermore, vitamin B_12_ inhibited α-synuclein fibrillation and disassembled pre-existing fibrils, reducing cytotoxicity. These findings show that vitamin B_12_ is a promising nutritional source that might be investigated as a novel functional food ingredient for the treatment of PD [43].

A brief overview of the possible mechanistic role of vitamin B in amelioration of PD associated pathophysiological changes have been illustrated in Figure 2.

### 3.3. Vitamin C

Vitamin C, also called ascorbic acid, is commonly found in citrus fruits such as oranges, lemons, and grapes and vegetables such as broccoli. It has antioxidant, anti-viral, anti-microbial, and anti-inflammatory properties [50]. Vitamin C plays a crucial role in the antioxidant system, neurotransmission modulation, synaptic potentiation, and myelination in the CNS [51]. Neurological disorders, such as atypical Parkinsonism, have a strong correlation with vitamin C deficiency. Clinical studies have shown that patients with IPD and vascular Parkinsonism are deficient in vitamin C [44,52]. A deficiency of vitamin C is directly associated with the risk of PD. PD patients have reduced levels of vitamin C in their plasma compared to control subjects, suggesting vitamin C is a potent biomarker for PD [8]. In line with this, some other studies have shown that dietary intake of antioxidants, including vitamin C, may reduce the risk of PD and slow the progression of Parkinsonian symptoms in older individuals [4,33,36]. Further, supplementation of vitamin C has also been shown to reduce protein oxidation, suppress H_2_O_2_ production, and increase anti-oxidant enzymatic activity in the DJ-1β mutant fly model of PD [53]. Mechanistically, vitamin C becomes internalized in the cell via a transporter called sodium-dependent vitamin C transporter type 2 (SVCT2). Inside the cell, vitamin C induces activation of nuclear factor erythroid 2-related factor 2/kelch-like ECH-associated protein 1(NRF2/Keap1) signaling, resulting in increased levels of antioxidant enzymes, thus aiding in the amelioration of oxidative stress-mediated PD progression [54,55]. Moreover, a study on MPTP-induced mice model of PD highlights that vitamin C also suppresses neuroinflammatory responses related to microglia and astrocytes’ activation and modulates TLR/NF-κB/NLRP3/IL-1β pathway, thus ameliorating PD-associated neuroinflammation [56,57]. Another in vivo study has shown that vitamin C inhibits MPP+-induced oxidative stress, preventing DA-ergic neuronal loss in PD [58]. Vitamin C also inhibits LB formation in PD, inhibiting apoptotic signaling-mediated cell death [59]. These shreds of evidence suggest that vitamin C, an antioxidant and anti-inflammatory agent, could be a potential therapeutic and preventive strategy against DA-ergic neurodegeneration in PD. A brief overview of the possible mechanistic role of vitamin C in amelioration of PD associated pathophysiological changes have been illustrated in Figure 3.

### 3.4. Vitamin D

Vitamin D is a steroid hormone that is essential for the functioning of body’s organs, including the brain [60]. Vitamin D can be obtained through food or produced in the skin by sunlight exposure. Almost most of the population across the globe receives their vitamin D demands from solar radiation. When it comes to vitamin D and geographical location, various elements such as ozone, latitude, clouds, month of the year, and season all have an impact on UV radiations at a particular geographical site. Deficiencies and insufficiencies in vitamin D and calcium have been found in several Asian nations, including India, China, Korea, and Japan, indicating the necessity of vitamin D for bone mass preservation and development. Due to geographical variations and the extent of exposure to UV radiation, there are variations in vitamin D synthesis in the population [61,62]. Insufficiency of vitamin D is frequent among the elderly worldwide and is the most common health issue associated with neurodegenerative disorders such as AD and PD [63]. Numerous studies have shown that PD patients had low serum vitamin D levels, suggesting that a deficiency of vitamin D is associated with an increased risk of developing PD [63,64].

Vitamin D deficiency has been attributed to poor memory, impaired verbal fluency, increased postural instability, and motor severity [65]. It has been demonstrated that PD patients, regardless of gender, have vitamin D deficiencies and, as a result, lower bone mineral density. Since vitamin D plays a crucial role in the metabolism of bones, its deficiency may cause an increased risk of falls and fractures. Consequently, there is an increased chance of fatal disability in PD patients [64].

The biological effect of vitamin D is regulated by vitamin D receptor (VDR). VDR is expressed in various brain regions, such as the cortex, caudate putamen, amygdala, hypothalamus, and DA-ergic neurons of SNpc. According to various reports, the VDR gene deletion results in motor dysfunction [66,67]. Furthermore, in the Asian population, the gene polymorphism rs1544410 has been related to PD susceptibility. A meta-analysis study has observed that deficiency and insufficiency of (25(OH)D), the biologically inactive form of vitamin D, and decreased exposure to sunlight are significantly associated with an enhanced risk of PD [68]. Along with the motor symptoms, vitamin D is also significantly associated with some non-motor symptoms in PD patients [64]. In elderly adults with MCI, vitamin D deprivation results in decreased hippocampus subfield volume and connectivity deficits. Consequently, it may result in the exacerbation of cognitive dysfunction [17]. Additionally, it has been demonstrated that Folk, an active VDR polymorphism, is also associated with loss of cognitive function in PD [69]. An independent association has also been demonstrated between vitamin D_3_ deficiency and olfactory dysfunction in PD patients [70].

Dietary supplementation with vitamin D_3_, the biologically active form, has demonstrated remarkable therapeutic value in PD. Vitamin D_3_ has been shown to improve behavioral impairments, reduce oxidative stress, and mitigate the loss of DA-ergic neurons and DA depletion in various animal models of PD [60,71,72]. In terms of molecular mechanism, it causes an increase in the expression of dopamine transporter and enzyme tyrosine hydroxylase. It suppresses ROS generation mediated by NADPH oxidase (NOX), MAO-B and inducible nitric oxide synthase (iNOS). Furthermore, vitamin D treatment reduces pro-inflammatory responses while activating anti-inflammatory responses, resulting in PD neuroprotection. Mechanistically, vitamin D_3_ decreases neuroinflammatory TNF-α, TLR-4, iNOS, CD11b, MAO-B, IL-I β, p47phox, and M1 microglia (pro-inflammatory) activation while promoting M2 microglia activation for an anti-inflammatory response. Moreover, vitamin D_3_ supplementation restores levels of brain-derived neurotrophic factor (BDNF) and glial-derived neurotrophic factor (GDNF) in the animal models of PD, thereby contributing to the survival of neurons [60]. GDNF binds to GDNF family receptor alpha 1 and then interacts with proto-oncogene tyrosine-protein kinase receptor Ret, forming a complex. This complex induces activation of intracellular signaling that confers neuronal survival [63]. GDNF, a powerful anti-oxidant, also stimulates the production of glutathione (GSH), SOD, and CAT in the striatum and aids in the regeneration of DA-ergic neurons [60]. Taken together, vitamin D holds immense antioxidant, anti-inflammatory, and neuroprotective potential in PD pathogenesis, which needs to be explored in more detail. Figure 4 depicts a brief overview of the probable molecular function of vitamin D in the amelioration of PD-related pathophysiological alterations.

### 3.5. Vitamin E

Vitamin E consists of a family of major lipid-soluble antioxidants that protect the cell membrane from polyunsaturated fatty acid-generated free radicals. It contains many lipophilic molecules (α-, β-, γ-, δ-tocotrienols, and α-, β-, γ-, δ-tocopherol). Vitamin E has an antioxidant effect and free radical scavenging properties and can prevent neuronal damage [73]. Patients with IPD and vascular Parkinsonism have shown deficiency in vitamin E, suggesting its potential link with neurodegenerative diseases, such as PD [44,52]. An in vivo study on a 6-OHDA-induced rat model of PD has shown that mice with vitamin E deficiency, when kept on a vitamin E-free diet for 52 weeks, showed a reduction in TH-positive cells, suggesting that vitamin E deficiency may cause DA-ergic neurodegeneration. However, supplementation of vitamin E provides neuroprotection by stabilizing cell membranes against the detrimental effects of lipid peroxidation (LPO) and scavenging free radicals produced by the metabolism of 6-OHDA in PD [18]. Dietary intake of vitamin E-rich food is also associated with a reduced risk of PD in in-vivo studies. Treatment with tocopherol derivative at a dosage of 20 mg/kg was reported to improve motor coordination and locomotor activity, boost neurotransmitter and antioxidant levels, and reduce α-synuclein expression and inflammatory cytokines in a haloperidol-induced mice model of PD [74]. Similarly, vitamin E supplementation also ameliorated motor deficits, improved biochemical oxidative stress biomarkers such as GSH and SOD levels, and significantly reduced LPO in the rotenone-induced rat model of PD [75]. Mechanistically, vitamin E becomes internalized inside the cell via LDL receptor-related protein (LRP) present on the neuronal surface. By controlling NRF2 and NF-κB signaling inside the cell, vitamin E lowers the burden of oxidative stress in the presence of PD [76]. Vitamin E also activates the estrogen receptor β/phosphatidyl inositol 3-phosphate kinase/Akt (ERβ/PI3K/Akt) signaling and inhibits the GSK-3β signaling to reduce neuroinflammation and hence suppress DA-ergic neurodegeneration [73,77]. Moreover, treatment with vitamin E also decreases the number of activated astrocytes in rat memory model of PD [76].

A recent meta-analysis study attempted to determine the effect of vitamin E on the risk of development of PD. It was reported that high vitamin E intake was associated with a lower risk of PD compared to the low vitamin E intake group [77]. Contrary to this finding, another study has shown that treatment with high vitamin E in combination with vitamin C delayed the progression of PD by 2.5 years in comparison to the placebo group [78]. A randomized, double-blind, placebo-controlled clinical trial involving 60 PD patients was conducted to evaluate the effect of co-supplementation of vitamin E (400 IU/day) and omega-3 fatty acid (1000 mg/day) on metabolic status and clinical symptoms of PD patients for 12 weeks. It was observed that co-supplementation of vitamin E and omega-3 fatty acid significantly improved UPDRS, increased total antioxidant capacity, GSH, and decreased high-sensitive C-reactive protein (hs-CRP) in comparison to the placebo group [79]. Thus, these findings suggest that vitamin E intake, alone or in combination with other antioxidative compounds, is neuroprotective against PD-induced neurodegeneration. A brief overview of the possible mechanistic role of vitamin E in amelioration of PD associated pathophysiological changes have been illustrated in Figure 5.

### 3.6. Vitamin K

Vitamin K is a group of fat-soluble vitamins existing in two forms, viz., vitamin K_1_ and K_2_. Vitamin K is a cofactor in synthesizing sphingolipids, the vital parts of the neuronal membranes. Epidemiological studies determining the relationship between vitamin K and PD are scarce. A case-controlled study involving 93 PD patients and 95 control subjects revealed that deficiency of vitamin K_2_ is associated with PD progression. Thus, vitamin K_2_ could be a potential biomarker for diagnosing PD [80]. Vitamin K_2_ is a potential treatment for mitochondrial abnormalities, particularly in PD patients lacking PINK1 or parkin. Vitamin K_2_ is essential and sufficient for transporting electrons in drosophila mitochondria, leading to the repair of mitochondrial abnormalities using a PINK1 mutant model. Like ubiquinone, vitamin K_2_ has been shown to transport electrons, enabling drosophila mitochondria to produce adenosine triphosphate (ATP) efficiently. Vitamin K_2_ was even helpful in addressing systemic locomotion abnormalities in adult PINK1 and parkin mutant flies [81]. In light of this, vitamin K_2_ has been recommended as a potential therapy option for mitochondrial dysfunction, particularly in PD patients with a PINK1 or parkin deficit. It has also been shown that vitamin K reduced α-synuclein fibrillization at sub-stoichiometric dosages [82]. A study by Yu et al. revealed that menaquinone-4 (*MK-4),* a vitamin K_2_ homolog, suppresses microglial activation in rotenone-treated BV2 cells by restoring mitochondrial membrane potential, reducing ROS production, and inhibiting NF-κB activation. MK-4 also prevented microglial-induced neuronal cell death, thus demonstrating the inflammation regulatory role of vitamin K_2_ in PD pathogenesis [83]. Before establishing a causative relationship between vitamin K_2_ and the onset of PD, nevertheless, further information and in-depth analyses are required. A brief overview of the possible mechanistic role of vitamin K in ameliorating of PD-associated pathophysiological changes has been illustrated in Figure 6.

A detailed account of the clinical studies associating the role of various vitamins in PD pathogenies has been summarized in Table 1. 

## 4. Conclusions and Future Directions

The neuropathological and behavioral deficits encountered in familial and sporadic PD can be primarily attributed to an upsurge in oxidative stress-related changes and neuroinflammation in the brain’s cellular milieu. To date, no effective treatment strategy against PD has been developed. However, the risk of PD can be prevented or delayed by adopting a healthy nutritional diet. A deficiency of various vitamins, including vitamins A, B, C, D, E, and K, has been indicated as a vital risk factor for PD onset. Thus, adequate intake of these vitamins can be considered a critical preventive measure against PD. Previous pre-clinical and clinical studies have demonstrated that dietary supplementation of various vitamins can reduce the prevalence of PD and improve motor deficits and clinicopathological changes associated with PD in the general population. The antioxidative role of vitamins and their biological role in influencing various gene expressions associated with PD may be advantageous in treating PD. Even though mitochondrial dysfunction and associated oxidative stress form the most critical link in mediating PD pathogenesis, the anti-oxidative potential of vitamins remains yet to be harnessed extensively for countering PD. Balanced healthy vitamin nutrition seems promising, although vitamin dose that may cause toxicity remains to be explored. The role of vitamins in related Parkinsonian syndromes is scarce and may require future studies for more insights into it. Dietary supplementation of vitamins regulates various signaling pathways, however detailed role of vitamin deficiency in contributing to PD pathology warrants further study in different experimental models and in clinical studies. Identification of molecules and associated signaling pathways through proteomic approaches could widen the horizon of knowledge for developing vitamin supplementation-based therapy against PD. Moreover, the epigenetic and other related mechanistic bases of vitamins in modulating various cellular functions must be decoded to understand the contribution of vitamins as an effective preventive and therapeutic option. Thus, more studies on understanding the neuroprotective vitamins and their active metabolites against PD are required in the future to verify vitamins as effective therapeutic options against PD.

## Figures and Tables

**Figure 1 brainsci-13-00272-f001:**
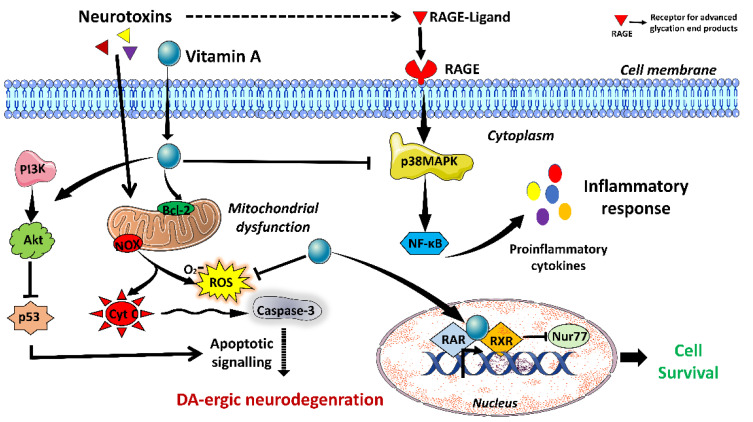
Possible neuroprotective mechanism of Vitamin A (Retinol) against Parkinson’s disease (**PD).** Retinoic acid (RA), the active form of vitamin A, has been shown to counter various PD-associated pathological changes. Following its entry inside the cell, retinoic acid activates PI3K/Akt signaling, which in turn inhibits the p53-mediated apoptotic process. RA also activates the antiapoptotic molecule Bcl-2. In addition, it inhibits receptor-advanced glycation-end receptor-advanced glycation-end (RAGE) products, key regulators of p38 mitogen-activated protein kinase/nuclear factor kappa B (p38MAPK/NF-κB) signaling mediated inflammatory response. Retinoic acid also exhibits reactive oxidative stress (ROS) scavenging ability as its presence decreases oxidative stress levels. Retinoic acid receptors, namely, retinoic acid receptor (RAR) and retinoid X receptor (RXR), located inside the nucleus, binds with retinoic acid to transcribe genes related to cell differentiation, and survival. RA enters inside the nucleus and binds to its receptors, inducing their transcription factor activity resulting in the expression of genes related to cell differentiation, and survival. RXR also modulates the export of Nur77 from the nucleus to the cytoplasm, resulting in the inhibition of Nur77-mediated apoptosis.

**Figure 2 brainsci-13-00272-f002:**
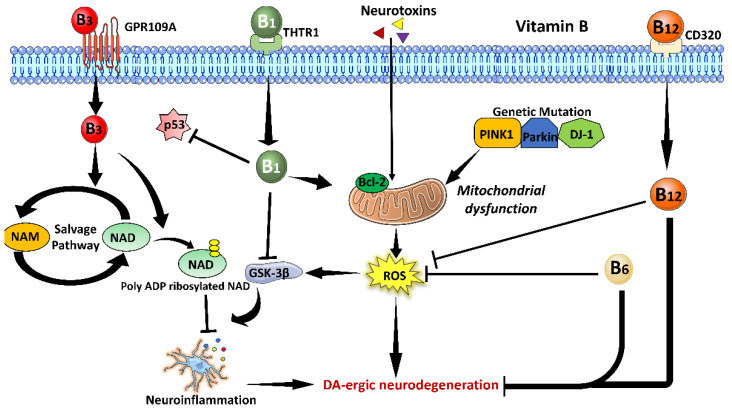
Possible neuroprotective mechanism of Vitamin B against Parkinson’s disease (**PD**). Genetic mutation in familial PD genes such as PTEN induced putative kinase 1 (PINK1), parkin, and DJ1 or environmental neurotoxins causes mitochondrial dysfunction resulting in reactive oxygen species (ROS) production. Increased ROS activates glycogen synthase (GSK-3β) followed by microglia activation-mediated neuroinflammation, which finally causes DA-ergic neurodegeneration. Vitamin B_1_ binds to its receptor THTR1 and enters inside the cell. It inhibits GSK-3β resulting in the inhibition of neuroinflammation. In addition, it inhibits p53 signaling and activates antiapoptotic Bcl-2 molecule resulting in the inhibition of apoptosis. Vitamin B_3_ enters the cell through its receptor GPR109A and increases biosynthesis of nicotinamide adenine dinucleotide (NAD) by salvage pathway. Additionally, it causes poly ADP ribosylation of NAD, which results in an anti-inflammatory response and thus, protection from DA-ergic neurodegeneration. Vitamin B_6_ also acts as a free radical scavenger molecule, decreases oxidative stress, and inhibits DA-ergic neurodegeneration. Vitamin B_12_ binds to its receptor CD320 and enters inside the cell. It acts as a free radical scavenger molecule, ameliorates oxidative stress, and inhibits DA-ergic neurodegeneration.

**Figure 3 brainsci-13-00272-f003:**
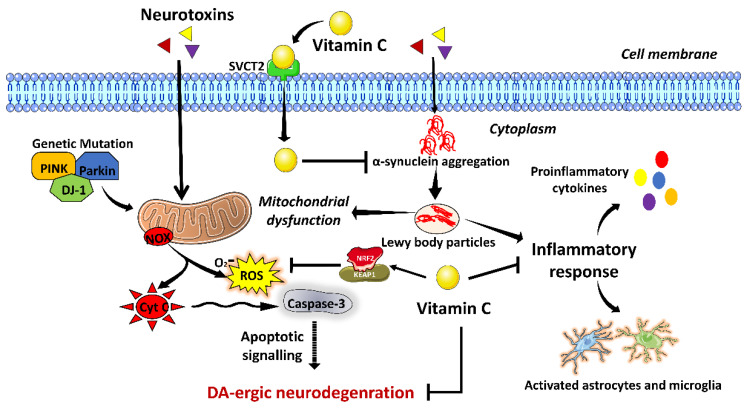
Possible neuroprotective mechanism of Vitamin C against Parkinson’s disease (**PD**). Along with mitochondrial dysfunction, PD-related neurotoxins also cause aggregation of α-synuclein protein in the form of Lewy bodies (LB). LB formation generates an inflammatory response and further exacerbates mitochondrial dysfunction. Vitamin C binds to its receptor SVCT2 and enters the cell. It inhibits α-synuclein aggregation and microglia activation-associated inflammatory response. Additionally, it activates nuclear factor erythroid 2-related factor 2/kelch-like ECH-associated protein 1 (NRF2/Keap1) signaling, resulting in the inhibition of reactive oxygen species (ROS), thus facilitating inhibition of DA-ergic neurodegeneration.

**Figure 4 brainsci-13-00272-f004:**
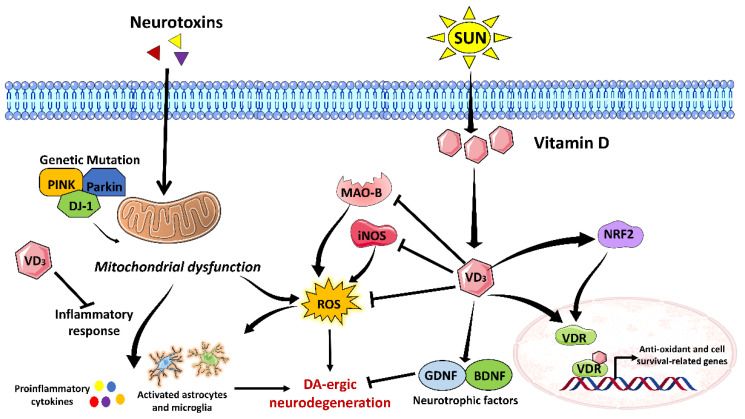
Possible neuroprotective mechanism of Vitamin D against Parkinson’s disease (**PD**). Vitamin D enters inside the cell and, being a free radical scavenger, reduces PD-associated reactive oxygen species (ROS) levels. Additionally, it also inhibits monoamine oxidase B (MAO-B) and inducible nitric oxide synthase (iNOS)-mediated ROS generation. The binding of vitamin D with its receptor VDR induces the transcription of antioxidants and cell survival-related genes. It also activates and induces the translocation of nuclear factor erythroid 2-related factor 2 (NRF2), from the cytosol to the nucleus. NRF2 induces transcription of antioxidant genes, such as glutathione (GSH), superoxide dismutase (SOD), and catalase (CAT), ameliorating oxidative stress. Vitamin D inhibits inflammatory response as well. Vitamin D also activates neurotrophic factors, such as brain derived neurotrophic factor (BDNF) and glial cell derived neurotrophic factor (GDNF), which confers protection to the neurons and inhibit DA-ergic neurodegeneration.

**Figure 5 brainsci-13-00272-f005:**
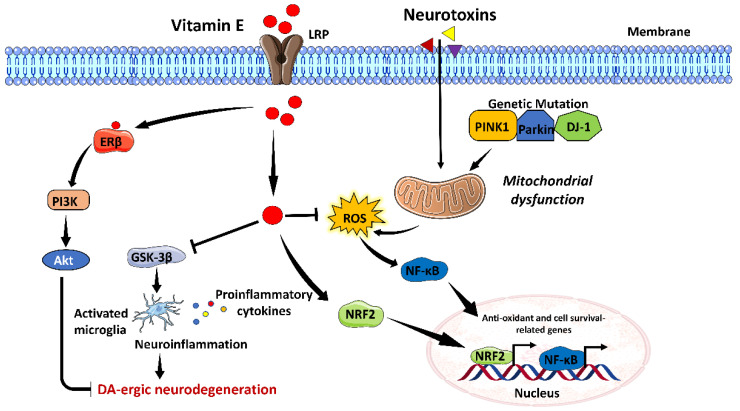
Possible neuroprotective mechanism of Vitamin E against Parkinson’s disease (**PD**). Vitamin E binds to its receptor LRP and enters the cell. It induces activation of nuclear factor erythroid 2-related factor 2 (NRF2), resulting in the transcription of antioxidants and cell survival-related genes during PD. Additionally, vitamin E scavenges ROS, resulting in the amelioration of oxidative stress. It also inhibits glycogen synthase kinase (GSK-3β) signaling-mediated neuroinflammation under a PD background. By binding to estrogen receptor (ER)-β, it activates PI3K/Akt signaling, resulting in the inhibition of DA-ergic neurodegeneration.

**Figure 6 brainsci-13-00272-f006:**
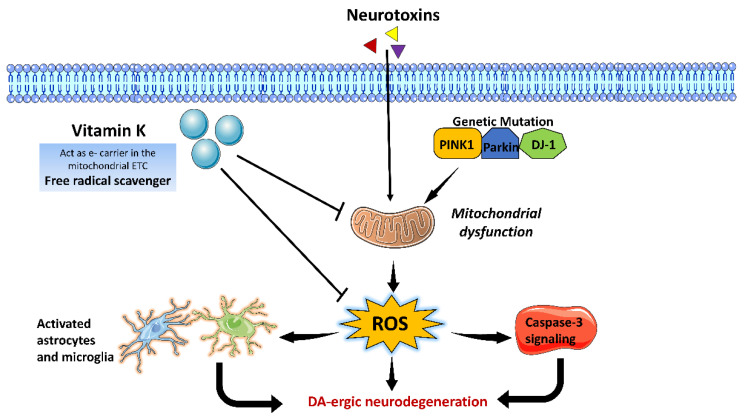
Possible neuroprotective mechanism of Vitamin K against Parkinson’s disease (**PD**). Vitamin K acts as a free radical scavenger and inhibit ROS during PD pathogenesis. It acts as an electron carrier in the mitochondrial electron transport chain (ETC) and regulates ATP synthesis.

**Table 1 brainsci-13-00272-t001:** Clinical studies investigating the association between vitamins and PD progression.

Vitamin	Study Design	Sample Size	Results	Vitamin and PD Association	Ref.
Vitamin A	Cohort-study	42 PD patients and 42 healthy controls	No significant difference in the serum levels of vitamin A and retinol-binding protein between the control and PD group	No association between vitamin A and the risk of PD	[24]
	Population-based cohort study	63,257 men and women, including 544 patients with incident PD	No precise dose-dependent association between dietary intake of vitamin A, E, and C and the risk of PD	No association between vitamin A and the risk of PD	[25]
Vitamin B_1_	Case-study	5 PD patients	Intake of daily 100–200 mg doses of parenteral thiamine improved movement, arm swings, and tremors in thiamine deficient PD patients	TD is associated with an increased risk of PD, and its supplementation may be beneficial	[27]
	Case-study	3 PD patients	A high dose of thiamine intake significantly improved motor coordination-related UPDRS, ranging from 31.3% to 77.3%	A high dose of thiamine intake is associated with improvement in PD symptoms	[28]
	Case-controlled study	96 PD patients and 375 control subjects	Deficiency of thiamine and folate caused olfactory dysfunction in PD patients	TD is associated with an increased risk of PD	[29]
	Case-controlled study	75 PD patients and 24 control subjects	In male PD patients, higher levels of phosphate and thiamine concentration, as well as higher MNA-total score, were correlated with a lower risk of MCI	Thiamine insufficiency and low phosphate levels increase the risk for PD-associated cognitive deficits	[30]
Vitamin B_3_	Case-controlled study	46 PD patients	PD patients with vitamin B_3_ deficiency were associated with GPR109A-mediated inflammation. Supplementation with 100 mg and 200 mg doses showed ameliorative effect	Vitamin B_3_ deficiency is associated with an increased risk of PD.	[38]
Vitamin B_6_	Case-controlled study	249 PD patients and 368 control subjects	Low consumption of vitamin B_6_ was associated with an elevated risk of PD	A deficiency of vitamin B_6_ increases the risk of PD	[51]
	Case-study	83-year-old woman with hypertension, coronary artery disease, and PD	Early detection and treatment of pyridoxine deficiency may reduce new-onset epileptic seizures and status epilepticus in PD patients	Vitamin B_6_ deficiency is associated with an increased risk of PD	[42]
	Population based cohort-study	5289 total participants, including 72 patients with incident PD	No association between dietary folate and vitamin B_12_ and the risk of PD. Vitamin B_6_ decreased the risk of PD.	Dietary vitamin B_6_ correlated with reduced risk of PD	[19]
Vitamin B_12_	Population-based cohort study	NA	Higher serum vitamin B_12_ at baseline level of PD diagnosis was correlated with a reduced risk of dementia	Vitamin B_12_ is associated with decreased risk of PD	[46]
	Longitudinal cohort- study	1741 participants	A low hazard ratio in subjects taking vitamin B_12_ + MVI and MVI groups for developing sensory symptoms of PD	Vitamin B_12_ is associated with a reduced risk of PD	[47]
Vitamin C	Cohort-study	75 PD patients and 75 healthy subjects	Patients with PD had considerably increased nitrite oxide and peroxynitrite but low vitamin C levels in the serum	Vitamin C deficiency is associated with an increased risk of PD	[8]
Vitamin D	Cohort-study	182 PD patients and 185 control subjects	PD patients had lower serum levels of 25 (OH)D than healthy controls	Vitamin D deficiency is associated with an increased risk of PD	[64]
		Patients with MCI were categorized as serum 25 (OH)D deficient (n = 27) or not deficient (n = 29) based on serum 25 (OH)D levels.	In older persons with MCI, low vitamin D levels were related with lower volumes of hippocampus subfields and connection impairments, which aggravated neurocognitive results.	Low vitamin D is associated with progression from MCI to major cognitive disorders.	[17]
	Observational study	145 PD patients and 94 control subjects	PD patients had lower serum levels of 25 (OH)D than healthy controls at baseline and at 18th-month follow-up session	A deficiency of 25 (OH)D is associated with increased motor severity and risk of bone fracture in PD patients	[65]
	Meta-analysis	NA	Significant associations between rs2228570 and PD risk were found in allelic, dominant, and additive models but not in the recessive model.	VDR polymorphism is associated with an increased risk of PD	[67]
	Meta-analysis	NA	Both 25 (OH)D insufficiency and deficiency were correlated with an increased risk of PD. However, vitamin D supplementation did not improve motor symptoms in PD patients	Deficiency of 25 (OH)D and reduced exposure to sunlight is associated with an increased risk of PD	[68]
	Observational study	39 drug-naive, de novo PD patients	Vitamin D was involved in the etiology of olfactory impairment in PD	Vitamin D deficiency increases the risk of olfactory dysfunction	[70]
Vitamin E	Meta-analysis	NA	High vitamin E consumption considerably decreased the chance of developing PD	High vitamin E intake is associated with a reduced risk of PD	[77]
	Randomized double-blind placebo-controlled study	60 PD patients	Co-supplementation with omega-3 fatty acids and vitamin E improved UPDRS in persons with PD	Vitamin E supplementation is associated with decreased risk of PD in older adults	[79]
Vitamin K	Case-controlled study	93 PD patients and 95 healthy controls	PD patients were deficient in serum vitamin K_2_ level	Vitamin K_2_ deficiency is associated with an increased risk of PD	[80]

## Data Availability

Not applicable.

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
