# Peer review of "Could Vitamins Have a Positive Impact on the Treatment of Parkinson’s Disease?"

_brainsci, 2023, doi:10.3390/brainsci13020272_

Round 1

Reviewer 1 Report

The issue of the role of vitamins in the pathogenesis of parkinsonisms is relatively poorly described in the literature. The work brings a quite interesting overview however several points could be further improved:

1. The introduction should also acknowledge the inflammation as a significant point in the pathogenesis of PD. The pathogenesis should be acknowledged also in the peripheral context. In this context authors should also refer to related disorders as atypical parkinsonisms - Ref.

Platelet-to-lymphocyte ratio and neutrophil-tolymphocyte ratio may reflect differences in PD and MSA-P neuroinflammation patterns. Neurol Neurochir Pol. 2022;56(2):148-155. doi: 10.5603/PJNNS.a2022.0014. Epub 2022 Feb 4. PMID: 35118638.

Inflammation in Parkinson's disease. Adv Neurol. 2001;86:83-9. PMID: 11554012.

Neutrophil-to-lymphocyte ratio (NLR) at boundaries of Progressive Supranuclear Palsy Syndrome (PSPS) and Corticobasal Syndrome (CBS). Neurol Neurochir Pol. 2021;55(1):97-101. doi: 10.5603/PJNNS.a2020.0097. Epub 2020 Dec 14. PMID: 33315235.

In the overview concerning vitamins it would be important to discuss the role of vitamins also in other parkinsonian syndromes.

2. Authors should also refer to the role of diet in the context of pathogenesis.

3. In the description of vitamin D, authors could also refer to the geographic distribution.

4. The review lacks a table summarizing the outcome of the work.

5. It would be valuable to elaborate on future perspectives.

Author Response

Please find the  authors responses to reviewer 1 as attachment.

Reviewer 2 Report

Dear authors,

The paper that was given to me to review is very interesting. Vitamins are natural and essential substances that exert many biological activities and represent an inexhaustible source of therapeutic possibilities. On the other hand, neurodegeneration processes form a group of socially significant diseases whose treatment represents a therapeutic challenge.

To make your article even better, I have a few recommendations. I have also pointed out some inaccuracies and omissions, as follows.

·         I find too many repetitions of PD” abbreviation in the abstract text. I think it would be better to be rewritten.

·         I believe that the article is not focused on the oxidative stress only, so it is better to include information about neuroinflammation as a pathogenic mechanism of the neurodegeneration process. Moreover, the review presents data on the anti-neuroinflammatory potential of some of the vitamins.

·         I recommend the authors to cite a more recent source of clinical trial data than No. 5 (1992) on line 49

·         Please, indicate the meaning of the abbreviations of the following lines 173, 309, 311/333, 346, define RAGE in Fig. 1, line 388 (Fig. 6).

·         The main text of the article needs rewriting in the past tense to be grammatically correct. The problem is evident in sections 3.1., 3.2.2., 3.4., and 3.5.

·         Please, provide references about data on lines 84-86, 99-102, and 219-220.

·         Please, rewrite the sentence on lines 86-88 because of the repetition of “Vitamin A” and on 126-129 “Vitamin B family”

·         I recommend the authors to change the title of item 3.2. from "Vitamin B" to "Vitamin B family"

·         In the sentence on lines 172-174, a period has been placed instead of a comma.

·         The sentence on lines 202-204 about Fig. 2 should start on a new line because it summarizes information about all B vitamins

·         What about Vitamin B12? There is data available on its importance in Parkinson's disease too, for example McCarter et al., 2019; Dietiker et al., 2019. I recommend the authors to include the information in the review.

·         I consider it unnecessary, after the number of the source of the information has been indicated, to write the name in the text as well – lines 261, 337

·         In my opinion, it is necessary to standardize the spelling of the substances - with a capital or small letter, the vitamins indexes should be in subscript, and the spelling of PINK1 and Parkin should not differ in the different subsections of the article.

·         I draw the attention of the authors that the mitochondrion in the figures is not fully drawn, except for the one in Fig. 1 and 3.

·         About this sentence: “Dietary intake of vitamin E-rich food is also associated with a reduced risk of PD.” – please, specify is it about experimental conditions or it is about clinical data?

·         On line 335, references with the numbers 97 and 100 are indicated, and the article contains only 67

·         It would be good, in my opinion, if in the sentence of lines 335-336 the authors specify that the data were established in experimental rats (Ref. 60).

·         In the conclusion of the article, the authors state that the presented review covers preclinical and clinical data from the last decade (line 399), but 11 references are from a previous period (Ref. No. 5,9,11,12,16,17,26,34, 36,60,62). 

Review date

January 11, 2023

Author Response

Please find the authors responses to Reviewer 2 as attachment.

Round 2

Reviewer 2 Report

Dear authors,

Thank you for the answer. I sincerely believe the revision made and the substantial changes in the manuscript have improved the final appearance of your article.

I consider the proposed article suitable for publication in its present form.

28 Jan 2023
